# Immunogenic Death of Hepatocellular Carcinoma Cells in Mice Expressing Caspase-Resistant ROCK1 Is Not Replicated by ROCK Inhibitors

**DOI:** 10.3390/cancers14235943

**Published:** 2022-11-30

**Authors:** Gregory Naylor, Linda Julian, Steven Watson-Bryce, Margaret Mullin, Robert J. Nibbs, Michael F. Olson

**Affiliations:** 1Cancer Research UK Beatson Institute, Garscube Estate, Switchback Road, Glasgow G61 1BD, UK; 2Institute of Cancer Sciences, University of Glasgow, Glasgow G12 8QQ, UK; 3Electron Microscopy Facility, School of Life Sciences, University of Glasgow, Glasgow G12 8QQ, UK; 4Institute of Infection, Immunity and Inflammation, School of Life Sciences, University of Glasgow, Glasgow G12 8QQ, UK; 5Department of Chemistry and Biology, Toronto Metropolitan University, 661 University Avenue Suite 1105, Toronto, ON M5G 1M1, Canada

**Keywords:** liver cancer, apoptosis, kinase, liver

## Abstract

**Simple Summary:**

The efficacy of some cancer chemotherapies relies on their ability to induce immunogenic cell death (ICD) to activate the immune system. In addition, it has been suggested that treatments that provoke ICD could be used in combination with chemotherapies to improve their anti-cancer actions. Having determined that mutating the caspase cleavage site in ROCK1 promoted necrosis-like inflammatory cell death during acute chemically induced liver damage and in hepatocellular carcinomas, we examined whether similar effects would be induced by pharmacological ROCK inhibition. Although mice expressing the caspase-resistant non-cleavable ROCK1 (ROCK1nc) had higher levels of neutrophils and CD8^+^ T cells in hepatocellular carcinoma tumours than in ROCK1 wild-type mice, indicative of greater ICD, two independent pharmacological ROCK inhibitors did not induce similar patterns. As a result, there is unlikely to be a clinical benefit derived from using pharmacological ROCK inhibitors to provoke ICD with standard of care therapies to increase their therapeutic efficacy.

**Abstract:**

The morphological changes during apoptosis help facilitate “immunologically silent” cell death. Caspase cleavage of the ROCK1 kinase results in its activation, which drives the forceful contraction of apoptotic cells. We previously showed that when ROCK1 was mutated to render it caspase-resistant, there was greater liver damage and neutrophil recruitment after treatment with the hepatotoxin diethylnitrosamine (DEN). We now show that acute DEN-induced liver damage induced higher levels of pro-inflammatory cytokines/chemokines, indicative of immunogenic cell death (ICD), in mice expressing non-cleavable ROCK1 (ROCK1nc). Hepatocellular carcinoma (HCC) tumours in ROCK1nc mice had more neutrophils and CD8^+^ T cells relative to mice expressing wild-type ROCK1, indicating that spontaneous tumour cell death also was more immunogenic. Since ICD induction has been proposed to be tumour-suppressive, the effects of two distinct ROCK inhibitors on HCC tumours was examined. Both fasudil and AT13148 significantly decreased tumour numbers, areas and volumes, but neither resulted in greater numbers of neutrophils or CD8+ T cells to be recruited. In the context of acute DEN-induced liver damage, AT13148 inhibited the recruitment of dendritic, natural killer and CD8^+^ T cells to livers. These observations indicate that there is an important role for ROCK1 cleavage to limit immunogenic cell death, which was not replicated by systemic ROCK inhibitor administration. As a result, concomitant administration of ROCK inhibitors with cancer therapeutics would be unlikely to result in therapeutic benefit by inducing ICD to increase anti-tumour immune responses.

## 1. Introduction

Apoptotic cell death is associated with profound morphological changes including cell contraction, blebbing, nuclear disintegration and apoptotic body release [1]. In addition, apoptotic cells display a variety of “find me” and “eat me” signals that aid in the efficient recognition and clearance of cell corpses by phagocytic cells including macrophages [2,3,4]. These events reduce the risk of cellular contents being released by dying cells in order to limit the induction of acute inflammatory and adaptive immune responses over time [5]. In the absence of efficient clearance of apoptotic cells, immunogenic cell death (ICD) may result in which the release of damage-associated molecular patterns (DAMPs) and induction of cytokines can promote the recruitment and activation of antigen-presenting cells, and subsequent activation of cytotoxic T-lymphocyte (CTL) mediated adaptive immunity [6]. Although cancers typically employ several strategies to evade detection and/or destruction by the immune system, inducing tumour cell ICD can improve the efficacy of cancer therapeutic agents by promoting anti-tumour immunity [7,8]. Some anti-cancer agents act both by blocking essential targets in tumour cells and by inducing ICD [9]. In addition, the physical interventions of irradiation, photodynamic therapy and high hydrostatic pressure have all been shown to induce cancer cell ICD by promoting the release of DAMPs and the induction of pro-inflammatory cytokines [10]. Therefore, therapeutic agents that directly promote ICD could have beneficial effects for cancer treatment.

The biochemical processes induced by pro-apoptotic stimuli culminate in the activation of executioner caspases that cleave substrate proteins at specific target sequences [11]. We and others discovered that caspase cleavage of the ROCK1 protein kinase at a highly conserved DETD sequence resulted in the generation of a constitutively-active fragment [12,13]. Caspase-mediated ROCK1 activation led to increased myosin light chain (MLC) phosphorylation and actin-myosin contraction that were necessary for rapid cell contraction, membrane blebbing and nuclear fragmentation during apoptosis [12,14,15]. Pharmacological inhibition of ROCK activity in apoptotic cells was sufficient to block increased MLC phosphorylation, membrane blebbing and nuclear disruption without affecting biochemical aspects of apoptotic cell death including caspase activation, DNA fragmentation, cytochrome C release and phosphatidylserine externalization [12,14,16]. A genetically modified mouse model in which a single amino acid change in the DETD caspase cleavage site that rendered ROCK1 insensitive to caspase cleavage revealed the obligatory role of ROCK1 cleavage and activation for MLC phosphorylation, the generation of cellular contractile force and the morphological aspects of apoptotic cell death [15]. Treatment of mice expressing the non-cleavable ROCK1 (ROCK1nc) with the liver selective hepatotoxin diethylnitrosamine (DEN) [17] that undergoes CYP2E1-mediated oxidation to become an active genotoxin [18] resulted in greater acute liver damage and neutrophil recruitment relative to mice expressing wild-type ROCK1 (ROCK1wt) [15]. Apoptotic ROCK1nc cells released more of the DAMP protein high mobility group B1 (HMGB1) than ROCK1wt cells, and DEN-induced acute liver damage and neutrophil recruitment could be significantly reduced by inhibiting HMGB1 or its cognate pattern recognition receptor Toll-like receptor 4 (TLR4) [15]. Taken together, these findings indicated that the inability of cells to undergo typical apoptotic morphological changes due to the absence of caspase-mediated ROCK1 activation induced a form of ICD.

DEN induction of hepatocyte apoptosis and tissue damage can progress to hepatocellular carcinoma (HCC) over time [17]. While both ROCK genotypes developed tumours 6 or 9 months after DEN treatment, there were fewer HCC tumours in ROCK1nc mice at both time points [15]. These observations suggested that the ROCK1nc mutation which affected apoptotic cell morphologies and resulted in acute hepatocyte ICD following DEN-treatment could also have a tumour-suppressive effect. Furthermore, these results raised the possibility that pharmacological ROCK inhibition could also have similar tumour suppressive effects by inducing ICD.

In this study, we observed that the ROCK1nc mutation altered the typical morphological changes of apoptotic hepatocytes in vitro, and resulted in greater induction of pro-inflammatory cytokines in ROCK1nc mice relative to ROCK1wt mice following acute DEN treatment, consistent with the absence of ROCK1 cleavage leading to ICD. DEN-induced HCC tumours in ROCK1nc mice displayed higher grade steatosis, which was accompanied by significantly more neutrophils and CD8^+^ T cells relative to ROCK1wt mice, while levels of Foxp3^+^ regulatory T cells (Tregs) were not different. Taken together, these results indicated that the ROCK1nc mutation induced immune responses that have been associated with tumour suppression and improved patient outcomes [7]. To determine if pharmacological ROCK inhibition could phenocopy the effect of the ROCK1nc mutation, wild-type ROCK1 mice were treated with the ROCK inhibitors fasudil [19] or AT13148 [20,21,22] following DEN-initiation of HCC. Although both ROCK inhibitors significantly reduced HCC tumour numbers, volumes and areas, neither increased neutrophil nor CD8^+^ T cell recruitment relative to vehicle controls. By examining the effect of AT13148 on acute DEN-induced liver damage, it was observed that the numbers of circulating platelets and lymphocytes were decreased, as were the proportion of dendritic, natural killer and CD8^+^ T cells recruited to damaged livers. Taken together, these observations indicate that the caspase-cleavage of ROCK1 is required to limit innate and adaptive immune responses to acute DEN-induced liver damage and spontaneous apoptosis in HCC tumours. However, potential therapeutic benefits from the tumour-suppressive actions of the ROCK1nc mutation could not be recapitulated by systemic pharmacological ROCK inhibition. As a result, it is unlikely that the addition of ROCK inhibitors as a cancer therapy adjuvant would improve therapeutic efficacy by inducing ICD and promoting anti-cancer immune responses. Nevertheless, ROCK inhibitor monotherapy did have a significant effect on reducing HCC tumour growth, suggesting that there may be clinical utility for ROCK inhibitors in the treatment of liver cancer, similar to the positive effects observed in pre-clinical studies for other cancer types including pancreatic ductal adenocarcinoma [22,23,24] and squamous cell carcinoma [25].

## 2. Materials and Methods

### 2.1. Mouse Model

ROCK1nc mice were generated at the Cancer Research UK Beatson Institute, and were bred and maintained as described in [15]. ROCK1wt and ROCK1nc mice were backcrossed for more than 10 generations into the C57Bl/6J strain. Mice were assigned to treatment groups by researchers who were not blinded with respect to genotypes. For routine genotyping, all animals were ear notched at weaning and samples sent to Transnetyx (Cordova, TN, USA) genotyping service for analysis.

### 2.2. Isolation of Hepatocytes

Hepatocytes were isolated from male and female ROCK1nc and ROCK1wt mice by a collagenase digestion and Percoll gradient method. Isolated cells were seeded on collagen-coated plates. For coating plates with collagen (Corning, Corning, NY, USA, 354249), 50 mg/mL collagen was prepared in water containing glacial acetic acid (6 µL in 5 mL water). Plates were coated for 1 h after which excess liquid was aspirated and plates left to dry in a fume hood.

For hepatocyte isolation, mice were euthanized using increasing concentrations of CO_2_ followed by cervical dislocation before the thoracic and peritoneal cavities were opened. Livers were perfused with Liver Perfusion Media (Gibco, Waltham, MA, USA, 17701038) at 37 °C via the right atrium and out through the portal vein. Once the liver became pale, Perfusion Media was switched to Liver Digest Media (Gibco, 17703034). Digested livers were excised, gall bladders removed, and then livers were transferred to dishes containing Liver Perfusion Media. Livers were gently teased with forceps to release cells into solution. Cell suspensions were then filtered through 70 µm cell strainers to remove undigested tissue and connective tissue. Cells were centrifuged at 135× *g* for 1 min and resuspended in 6 mL of hepatocyte media (William’s E media (Gibco, 22551-022) with 3% fetal bovine serum (FBS; Gibco, 10270), 100 U/mL penicillin with 100 µg/mL streptomycin (Gibco, 15140-122) and 2 mM L-glutamine (Gibco, 25030-032).

Hepatocytes were separated using a three-layer discontinuous Percoll (Sigma, St. Louis, MI, USA, P1644) gradient. The Percoll density solutions used were; 1.12 g/mL, 1.08 g/mL and 1.06 g/mL. To make the 1.12 g/mL solution, 2.63 mL Percoll, 0.3 mL phosphate-buffered saline (PBS; 170 mM NaCl, 3.3 mM KCl, 1.8 mM Na_2_HPO_4_, 10.6 mM H_2_PO_4_) and 64.6 μL water were mixed; for the 1.08 g/mL solution, 2.86 mL Percoll, 0.5 mL PBS and 1.63 mL water were mixed; and for the 1.06 g/mL solution, 2.085 mL Percoll, 0.5 mL PBS and 2.42 mL water were mixed. Percoll gradients were generated by layering 3 mL of the 1.12 g/mL solution at the bottom over which 5 mL of the 1.08 g/mL solution was carefully layered followed by 5 mL of the 1.06 g/mL solution. Over this, 6 mL of cell suspension was carefully layered while avoiding gradient disturbance. The cell suspension was fractionated by centrifugation without brakes at 750× *g* for 20 min. After centrifugation, the layer of cells between 1.06 g/mL and 1.08 g/mL that contained live hepatocytes was collected and added to 10 mL of hepatocyte media. Cell suspensions were centrifuged at 135× *g* for 1 min. Cell pellets were then resuspended in hepatocyte media and seeded on collagen coated plates for analysis.

### 2.3. Time-Lapse Microscopy of Hepatocytes

ROCK1wt and ROCK1nc primary hepatocytes were plated on 6-well glass bottom dishes pre-coated with collagen. Cells were serum-starved overnight prior to induction of apoptosis with 1 µM ABT-199 (Selleckchem, Houston, TX, USA, S8048) diluted in serum-free media. Bright field time-lapse microscopy images were acquired with a 10× objective using a Nikon TE 2000 microscope with a heated stage and 5% CO_2_ gas line. Immediately after induction of apoptosis, dishes were transferred to the microscope, 3–4 random positions were chosen from each well and time lapse images were taken every 3 min for 18 h.

For apoptosis assays, images were obtained using the Incucyte Zoom time lapse microscope with post acquisition analysis performed on Incucyte Zoom software. To measure caspase activation, media containing 2 µM CellEvent Caspase-3/7 Green was added to wells along with indicated concentrations of AT13148 with subsequent imaging on the Incucyte Zoom time lapse microscope. To measure cell death, media containing 25 nM Sytox Green along with indicated concentrations of AT13148 was added to wells with subsequent imaging on the Incucyte Zoom time lapse microscope.

### 2.4. Scanning Electron Microscopy (SEM)

Hepatocytes were cultured on collagen coated 10 mm glass cover slips in 24 well plates. Apoptosis was induced using UVC radiation (254 nm wavelength at 0.3 J/cm^2^) and cells were processed for SEM at 24 h post apoptosis induction. Cover slips were washed twice in filtered PBS then fixed in 1.5% glutaraldehyde (Sigma, G5882) in 0.1 M sodium cacodylate for 1 h at room temperature. Samples were then washed twice in 0.1 M sodium cacodylate and stored in 0.1 M sodium cacodylate. Cells were washed with phosphate buffer (3 washes for 10 min), post-fixed in 1% Osmium Tetroxide/0.1 M Sodium Cacodylate buffer for 1 h then washed for 3 × 10 min in distilled water. Dehydration steps were through a graded ethanol series (30%, 50%, 70%, 90%) for 10 min each, then 100% ethanol 4 times for 5 min each. Samples were dried with hexamethyldisilazane (Sigma, H4875) before coating with gold/palladium, and were imaged on a JEOL 6400 Scanning Electron Microscope running at 10Kv. TIFF images were captured using Olympus Scandium software at the Electron Microscopy Facility at the University of Glasgow.

### 2.5. Preparation and Administration of Substances

To prepare diethylnitrosamine (Sigma, N0258), 100 mL of sterile PBS was injected into the bottle containing 1 g of DEN to give a final concentration of 10 mg/mL. The procedure was performed in the fume hood. For short-term studies assessing DEN-induced hepatic injury, mice were treated with a single dose of DEN at 100 mg/kg body weight by intraperitoneal injection.

To prepare fasudil, 1 g of fasudil (LC Laboratories, Woburn, MA, USA, F-4660) was dissolved in 50 mL distilled water giving a final concentration of 20 mg/mL. Fasudil was then aliquoted and stored at −20 °C until required for dosing. Mice were dosed with 100 µL (2 mg) fasudil twice daily for the duration of the study.

To prepare AT13148 (Selleckchem, S7563), compound was dissolved in 89% H_2_O/10% (*v*/*v*) DMSO (Sigma, D2650)/1% (*v*/*v*) Tween 20 (Sigma, P1379) at a concentration of 8 mg/mL. The dissolved drug was kept at 4 °C for a maximum of 2 weeks. Mice were dosed three times per week (Monday, Wednesday, Friday) for the duration of studies performed. Mice were dosed initially with 30 mg/kg for the first two weeks, then 40 mg/kg thereafter.

### 2.6. Tissue Collection and Fixation

Animals were euthanized by carbon dioxide inhalation, weights recorded, and blood collected for haematology or biochemistry analysis. Necropsy was performed immediately to prevent tissue autolysis. Tissues collected were fixed for at least 24 h by immersing in 10% neutral buffered formalin. For acute liver studies, the left lobe of the liver was routinely used for histological analysis and fixed in 10% neutral buffered formalin. Small pieces from the median lobe were frozen using dry ice or collected in RNALater for RNA analysis. The caudate lobe was frozen using dry ice for making cryosections.

For DEN-induced hepatocellular carcinoma studies, mice were examined for visible lesions in all organs, liver weights were recorded, and the number of tumours counted. Tumour foci were measured using digitised vernier callipers. Steatosis was graded on H&E-stained sections based on semi-quantitative estimation of percentage of steatotic hepatocytes, according to the following criteria: grade 0, less than 5%; grade 1, 5 to 25%; grade 2, 26 to 50%; grade 3, 51 to 75%; grade 4, more than 75%. Lungs were also collected to analyse metastatic lesions. Tissue were either fixed in 10% neutral buffered formalin for histological analysis or maintained in PBS until analysis. Fixed tissues were then processed, paraffin embedded and sectioned by the Cancer Research UK Beatson Institute Histology Service.

### 2.7. Histology

Paraffin-embedded tissue sections of 4 μm thickness were cut and routinely stained with haematoxylin and eosin (H&E) by the Beatson Histology Service. Blank paraffin sections were stored at 4 °C until ready for use. Frozen liver tissue was embedded in OCT and cut at 12 µm thickness and mounted on super-frost slides. Cryosections were stored at −80 °C until ready to be used.

### 2.8. RNA Isolation and Sequencing

RNA was isolated using the Qiagen RNasy Mini Kit according to the manufacturer’s instruction. RNA concentration and purity was determined using NanoDrop 2000 spectrophotometer. RNA quality was confirmed using the Agilent RNA Screen Tape assay and the Aligent 2200 TapeStation. Total RNA was used to generate an oligo dT-enriched library with the Illumina TruSeq RNA Library Preparation kit v2.0. Quality and quantity of the DNA library was assessed using an Agilent 2100 Bioanalyzer and a Qubit (Thermo Fisher Scientific, Waltham, MA, USA), respectively. The library was run on the Illumina NextSeq 500 platform using the High Output 75 cycles kit (2 × 36 cycles, paired-end reads, single index). RNAseq data were analysed by the Computational Biology department at the CRUK Beatson Institute. Gene set enrichment analysis (GSEA) was run using GSEA v4.1.0 for Windows [26].

Quality control checks of raw RNA-Seq data files were done with fastqc (http://www.bioinformatics.babraham.ac.uk/projects/fastqc/, accessed on 29 June 2016) and fastq_screen (http://www.bioinformatics.babraham.ac.uk/projects/fastq_screen/, accessed on 29 June 2016). RNA-Seq reads were aligned to the mouse genome (GRCm38.75) using TopHat2 [27] and BAM files were further processed with htseq_count (http://www.huber.embl.de/users/anders/HTSeq/doc/count.html, accessed on 29 June 2016). Differential analysis of count data was performed by the DESeq2 package [28].

### 2.9. Multiplex Cytokine Analysis

MILLIPLEX Multiplex Assays Using Luminex (Millipore, MCYTOMAG-70K) were used to measure the levels of 14 cytokines (IFN-γ, IL-1α, IL-1β, IL-2, IL-4, IL-5, IL-6, IL-10, IL12p40, IL12p70, CCL2, CCL3, CXCL9, TNFα). Serum levels of these cytokines were analysed using the Mouse Cytokine/Chemokine Magnetic Bead Panel Kit following manufacturer’s instructions. Briefly, 200 μL of wash buffer was added to each well of 96-well plates, which were sealed and mixed on a plate shaker for 10 min at room temperature after which the wash buffer was decanted. Inverting the plates and tapping them onto absorbent towels removed residual wash buffer. 25 µL of standards or controls were added to the appropriate wells and 25 µL of assay buffer was added to the sample wells. 25 µL of serum matrix was added to the background, control and standard wells. 25 μL of serum samples diluted 1:2 in assay buffer was added to the sample wells. All serum samples were measured in duplicate. 25 μL of premixed magnetic beads conjugated to antibodies for all 15 analytes were added to each well. Plates were sealed, protected from light and incubated overnight at 4 °C on a plate shaker. This was followed by a wash step (two times with 200 μL of wash buffer) on a magnetic plate washer. 25 µL of detection antibodies was added to each well, and plates were sealed and incubated with agitation for 1 hour at room temperature. Streptavidin-phycoerythrin (25 μL) was added to each well, and plates were sealed and incubated with agitation for another 30 min at room temperature, followed by a final wash step. Magnetic beads were then re-suspended in sheath fluid and assayed on a Luminex 200 analyser using xPOTENT software.

### 2.10. Immunohistochemistry

Immunohistochemistry was routinely performed on 4 µm thick formalin-fixed paraffin-embedded (FFPE) sections. Sections were deparaffinised in xylene (3 washes of 5 min each) and rehydrated through decreasing concentrations of ethanol (2 washes of 3 min each in 100% ethanol, 1 wash for 3 min in 70% ethanol). Slides were then washed in distilled water (dH_2_O). To unmask antigens, slides were boiled at 98 °C in a water bath (WB) for the specified time in antigen retrieval buffer. The buffers used varied with the antibody (Table 1). If Proteinase K (Dako, Jena, Germany, S2032) was used as the retrieval agent, then the slides were incubated at room temperature for the specified time (Table 1). Slides were left to cool down to room temperature for at least 30 min and then rinsed in distilled water. Endogenous peroxidase activity was quenched by incubating sections in 3% H_2_O_2_ (Fisher, 10687022) in PBS for 10 min. After slides were rinsed once in distilled water and once in Tris-buffered saline + Tween 20 (TBST; 20 mM Tris-HCL pH 7.5, 136 mM NaCl, 0.1% (*v*/*v*) Tween 20), sections were blocked using blocking solution containing 5% serum (depending on the species in which the secondary antibody was raised) in TBST for 30 min at room temperature. Primary antibody diluted in blocking solution was added to the slides and kept in a humidified chamber for the specified time. Primary antibody dilutions and incubation times are indicated in Table 1. Slides were rinsed three times in TBST to remove excess/unbound primary antibody. HRP-labelled secondary antibodies were added to the slides and incubated at room temperature for 30 min at room temperature. For secondary antibody staining, either ImmPRESS kits (anti-goat, MP-7405; anti-rat, MP-7444) or Envision+ Reagents (Dako: rabbit, K4002) were used. Slides were rinsed again three times in TBST before staining with 3,3′-diaminobenzidine tetrahydrochloride (DAB; Dako, K3467), as chromogen. Sections were incubated with DAB solution until brown colour develops and the slides were immediately transferred to TBST to stop the reaction.

For counterstaining, slides were rinsed in distilled water and then stained with haematoxylin for 1 min. After rinsing in distilled water, the slides were then transferred to Scots tap water for 1 min or until blue colour developed to the desired intensity. Slides were rinsed in distilled water and the sections were dehydrated using increasing concentrations of ethanol (1 wash for 3 min in 70% ethanol, 2 washes of 3 min each in 100% ethanol). Slides were then cleared by three washes in xylene for 3 min each, before mounting with DPX mountant (Sigma, 44581). In all staining experiments, omitting the primary antibody on one slide served as the negative control. Sections were analysed using a bright field microscope (Olympus BX51) and representative images were obtained at different magnifications.

### 2.11. Immunofluorescence Staining of Frozen Sections

Immunofluorescence staining of cryosections was performed on 12 µm thick cryosections. Tissue sections were fixed by immersing the slides in pre-cooled acetone (−20 °C) for 10 min. Slides were removed from fixative and acetone was allowed to evaporate from the tissue sections for 20 min. After slides were rinsed in PBS, cryosections were blocked in blocking buffer (5% serum in PBS) for 1 h at room temperature. Anti-CD8 primary antibody diluted in blocking buffer (1:100) was added to the slides and kept in a humidified chamber for 2 h at room temperature. Slides were rinsed twice in PBS before adding Alexa Fluor 594 conjugated goat anti-rat secondary antibody (ThermoFisher Scientific, A48264) diluted 1:500 in blocking buffer and incubated for 1 h at room temperature. Slides were then rinsed three times in PBS before mounting with Vectashield DAPI hard set (Vector, H-1500). Cryosections were analysed using a fluorescence microscope (Olympus BX51) and images were obtained at 20× magnification.

### 2.12. Hepatocellular Carcinoma Induction

Male mice were injected intraperitoneally (IP) with DEN as a single 40 μL dose (400 μg, equates to ~10 µL/g) at 14 days old. Mice were checked at regular intervals for general health and signs related to hepatocellular carcinoma such as hepatomegaly (swollen abdomen), pain (hunched, high gait), weight loss (pinched shoulders) or reduced mobility. The experimental end point was defined as 41 weeks, or in the event of development of clinical signs, at which point mice were humanely culled and samples collected. Mouse data such as body weight and organ weight were recorded, liver tumours were counted and measured using calipers, and lungs and spleen were assessed for metastases. Blood samples were additionally taken for analysis of haematological parameters and liver function tests. For FFPE samples, tissues were collected and placed directly into 10% neutral buffered formalin solution. Samples were placed on a rocker for 24 h then moved to a 70% ethanol solution until paraffin embedding. Tumour areas were calculated using the following formula; Area = π × (Length/2) × (Width/2). Tumour volumes were calculated with the following formula; Volume = (Length × width^2^)/2

### 2.13. Haematology

Mice were euthanized by carbon dioxide inhalation and blood samples were collected via cardiac puncture in potassium-EDTA tubes. Haematology analysis was performed by the Clinical Pathology Lab at the University of Glasgow Veterinary School.

### 2.14. Liver Function

Mice were euthanized by carbon dioxide inhalation and blood samples were collected via cardiac puncture in 1.5 mL microcentrifuge tubes. Blood was allowed to clot at room temperature for at least 30 min and then centrifuged at 1500× *g* for 15 min at 4 °C. The resulting supernatant (serum) was transferred to fresh tubes for analysis. If the serum samples were not analysed immediately, they were aliquoted and stored at −20 °C until ready for analysis. Liver function tests were performed by the Clinical Pathology Lab at the University of Glasgow Veterinary School, where serum concentrations of alanine transaminase, total bilirubin, alkaline phosphatase, aspartate aminotransferase, total protein, albumin and globulin were determined on an automated Siemens Dimension Xpand Plus Chemistry system using International Federation of Clinical Chemistry methodologies

### 2.15. Flow Cytometry

Cell suspensions were generated from mouse livers by first perfusing livers with PBS, then dissecting them and placing into ice cold PBS. The liver tissues were dissociated using scissors and incubated in a solution of 5 mL PBS containing 1 mg/mL collagenase D (Millipore, Burlington, MA, USA, 01803), 0.5 mg/mL dispase (Millipore, D4818) and 0.1 mg/mL collagenase P (Millipore, 11213857001) at 40 °C for 30 min shaking at 750 rpm. The cell suspensions were then diluted in 45 mL RPMI 1640 (Gibco, 31870-025) and passed through a 70 µM filter twice. The suspensions were then spun at 300× *g* for 5 min and resuspended in red blood cell (RBC) lysis buffer (Biolegend, San Diego, CA, USA, 420302) for 5 min. Following termination or RBC lysis in PBS, the suspensions were then spun down and resuspended in PBS.

2.5 × 10^6^ cells were stained in a volume of 100 µL unless otherwise stated. Cell suspensions were incubated in Zombie NIR live dead dye (Biolegend, 423106) at 1:250 in PBS at room temperature for 20 min. Suspensions were spun at 300× *g* for 3 min and washed twice in PBS. Suspensions were then incubated in Fc block at 1:250 in FACS buffer (PBS with 2% (*v*/*v*) FBS, 0.5% (*w*/*v*) sodium azide, 0.2 mM EDTA) for 20 min at room temperature then washed twice in FACS buffer. Cells were stained with antibodies for 30 min at room temperature (Table 2). Cells were then spun down and washed twice in FACS buffer. Secondary staining containing secondary antibodies and/or streptavidin fluorophore conjugate was then performed in FACS buffer at room temperature for 30 min. Cells were again spun down and washed twice in FACS buffer. Cells were then resuspended in FACS buffer and an equal volume 4% paraformaldehyde/PBS (final concentration 2%) was added for fixation.

Samples were analysed and data collected using an Attune NxT acoustic focusing flow cytometer. Post-acquisition analysis was performed using FlowJo v10 software. Separate staining panels were utilised to identify myeloid lineage leukocytes and lymphoid lineage leukocytes. Myeloid gating was performed as a variation of the strategy previously described in reference [29]. Briefly after exclusion of debris, doublets and dead cells, leukocytes were identified as CD45^+^. Further gating of the CD45^+^ populations identified immune cell types as follows:Neutrophil—Ly6G^+^ CD11b^+^Monocyte—Ly6G^−^, MHCII^-^ SCC^lo^, CD11b^+^ CD115^+^-Inflammatory Monocyte—Ly6C^hi^ CD11c^-^-Resident monocyte—Ly6C^−^ CD11c^+^Macrophage— Ly6G^−^, MHCII^+^ or SCC^hi^, CD11b^+^ F480^+^
-Recruited Macrophage—Clec4f^−^-Kupfer Cell—Clec4f^+^
Dendritic Cell—Ly6G^−^, MHCII^+^ or SCC^hi^, F480^−^, CD11c^+^
-Classical DC—PDCA1^−^, Siglec H^−^-Plasmacytoid DC—PDCA1^+^, Siglec H^+^


For lymphoid gating, following identification of CD45^+^ leukocytes as above, immune cell types were identified as follows;


T Cell—CD3^+^, CD19^−^-αβ T Cell—TCRβ^+^, TCRγδ^−^
oCD4 T Cell—CD4^+^, CD8^−^
▪T_reg_—FoxP3^+^oCD8 T Cell—CD4^−^, CD8^+^
-γδ T Cell—TCRβ^−^, TCRγδ^+^


B Cell—CD3^−^, CD19^+^, B220^+^Natural Killer—NK1.1^+^
-NK Cell—NK1.1^+^, CD3^−^-NKT Cell—NK1.1^+^, CD3^+^


### 2.16. Statistical Analysis

Graphpad Prism 9 software was used for statistical analysis of data and generation of graphs. The specific test used for analysis is stated in each figure legend.

## 3. Results

### 3.1. Caspase Cleavage of ROCK1 Affects Hepatocyte Morphology, Liver Gene Expression and Cytokine Release following Genotoxin Treatment

Julian et al. [15] previously showed that mouse embryonic fibroblasts (MEFs), homozygous for the ROCK1nc genetic modification, had less contractile force generation during apoptotic cell death relative to wild-type MEFs, leading to fewer apoptotic blebs. To determine if similar effects would be observed in hepatocytes, primary hepatocytes from ROCK1wt and ROCK1nc mice were isolated, treated with the apoptotic stimulus of the Bcl2 selective BH3 mimetic ABT-199 (1 µM), then imaged using bright field microscopy at 3 min time intervals. While ROCK1wt hepatocytes underwent rapid and forceful contraction that produced many blebs (Figure 1A, left panels; Appendix A), apoptotic ROCK1nc hepatocytes were less compact, had more residual contraction fibres and fewer apoptotic blebs (Figure 1A, right panels; Appendix A). Scanning electron microscopy revealed that the cell bodies of ROCK1WT hepatocytes were studded with large blebs, while ROCK1nc were essentially devoid of blebs and had large patches of membrane adjacent to the cell (Figure 1B). Therefore, the ROCK1nc mutation that renders ROCK1 resistant to caspase mediated proteolysis and activation similarly affected the apoptotic morphological features of contraction and blebbing for hepatocytes and fibroblasts [15].

The liver-selective pro-hepatotoxin diethylnitrosamine (DEN) [17] can be metabolized by CYP2E1 [18] to become genotoxic [30], resulting in hepatocyte apoptosis and tissue damage that can progress to cirrhosis and hepatocellular carcinoma (HCC) [17]. The toxic effects of DEN occur with high penetrance in male mice, with female mice being relatively insensitive [17]. After intraperitoneal (IP) injection of 10 week old male mice with 100 mg/kg DEN, haematoxylin and eosin (H&E) staining of livers isolated 72 h later revealed markedly greater levels of steatosis in ROCK1nc mice compared to livers from ROCK1wt mice (Figure 2A), consistent with previous observations [15]. RNA was isolated from 5 untreated ROCK1wt and ROCK1nc livers, and from 5 ROCK1wt and ROCK1nc livers 72 h after DEN treatment, and polyA^+^-enriched RNA was sequenced. Comparison of sequence reads from untreated ROCK1wt and ROCK1nc revealed that there were 571 genes differentially expressed by at least 1.5 fold (adjusted *p* < 0.05). Gene set enrichment analysis of the Reactome gene sets identified the gene set “Neutrophil degranulation” was significantly positively enriched (nominal *p* < 0.05) in ROCK1nc mice, and the gene set “Transport of small molecules” was significantly negatively enriched. In total, 2515 genes were significantly altered in expression by at least 1.5 fold (adjusted *p* < 0.05) between untreated and DEN treated ROCK1wt livers (Appendix A), and 1255 genes that were similarly altered in expression between untreated and DEN treated ROCK1nc livers (Appendix A), of which 740 were in common (Figure 2B). The direction of change was consistent for 96.5% of the 740 common genes (Appendix A), although the magnitude of positive and negative gene expression changes was significantly greater for ROCK1wt than for ROCK1nc livers (Appendix A). Gene set enrichment analysis revealed that there were four Reactome gene sets (out of 1604 total gene sets) [31,32] that were significantly (nominal *p* < 0.05) altered in both ROCK1wt and ROCK1nc livers (Figure 2C). In addition to the top hit “Transcriptional Regulation by tp53” gene set that likely reflects the DNA damaging actions of DEN [33], the next three gene sets were all related to immune signalling pathways.

To determine if there were differences in cytokine induction between ROCK1wt and ROCK1nc mice following DEN treatment, the levels of 14 cytokines/chemokines were measured by cytokine arrays. By comparing the fold-change induction of each cytokine/chemokine for vehicle versus DEN treatment for each genotype, there was significantly greater induction in ROCK1nc mice of interleukins (IL) IL-6, IL-1β, IL-10, tumour necrosis factor α (TNFα), IL-12-p40, IL-4, IL-2, IL-1α, and C-C motif chemokine ligand 2 (CCL2) (Figure 2D). The induction of IL-5, CCL3, IL12-p70, C-X-C motif chemokine ligand 9 (CXCL9) and interferon γ (IFNγ) was not different between ROCK1wt and ROCK1nc mice, and no cytokines/chemokines were induced to a significantly greater extent in ROCKwt mice (Figure 2D). Taken together, these observations indicate that ROCK1nc hepatocytes undergo aberrant morphological changes during apoptotic death, which leads to greater DEN-induced steatosis and inflammation.

### 3.2. Hepatocellular Carcinomas from ROCK1nc Mice Show Increased Neutrophil Infiltration, CD3+, and CD8+ T Cell Recruitment Compared to ROCK1wt Mice

Treatment of 14 day old male mice with 400 µg DEN by IP injection results in the development of HCC with high penetrance [15,17], with evidence of spontaneous apoptosis in tumours [34]. ROCK1nc mice were previously shown to develop fewer HCC tumours than ROCK1wt mice, suggesting that tumour development was affected by the inability of tumour cells to cleave and thereby activate ROCK1 [15]. Similar to there being more steatosis in ROCK1nc livers following acute DEN-treatment [15], there was a significantly (*p* < 0.001, Chi-square test) higher proportion of high grade (≥3) steatotic HCC tumours in ROCK1nc livers relative to ROCK1wt livers (Figure 3A; Appendix A). In addition, steatosis was detectable in liver tumours of 12/12 ROCKnc mice and only 10/12 ROCKwt mice. Despite the greater steatosis in ROCK1nc tumours, there were no significant difference in biochemical markers of liver function and health between ROCK1wt and ROCK1nc mice (Appendix A). Immunohistochemical (IHC) analysis revealed that there were significantly more S100A9^+^ neutrophils in ROCK1nc tumours relative to ROCK1wt tumours (Figure 3B), and no difference in F4/80^+^ macrophages between the genotypes (Figure 3C), comparable to the observations for acute DEN treatment [15]. Since previous studies have reported that low ratios of Foxp3^+^ regulatory T cells (Tregs) to either CD3^+^ or CD8^+^ T cells were correlated with better prognosis for patients with HCC [35,36,37,38], the levels of each cell types was assessed in ROCK1wt and ROCK1nc HCC tumours. Both CD3^+^ (Figure 4A) and CD8^+^ T cells (Figure 4B) were detected at significantly higher levels in ROCK1nc liver tumours than in ROCK1wt tumours, while there was no difference in Foxp3^+^ Treg numbers (Figure 4C). These results indicate that the enhanced pro-inflammatory cytokine responses observed in ROCK1nc mouse livers after acute DEN-treatment (Figure 2D) are paralleled by greater infiltration of neutrophils, CD3^+^ and CD8^+^ T cell in HCC tumours (Figure 3 and Figure 4).

### 3.3. Pharmacological ROCK Inhibition Reduces DEN-Induced Liver Tumour Numbers without Increasing Immune Cell Recruitment

A proposed cancer treatment strategy is to enable the immune system to more effectively target cancers by increasing the immunogenicity of tumour cells by transforming spontaneous or therapy-induced cell death from immunologically silent apoptosis to a more inflammatory necrotic-like form [5,39,40,41]. Results from the ROCK1nc mouse model indicate that blocking caspase-induced ROCK1 activation leads to increased acute inflammation (Figure 2D) that results in greater innate and adaptive immune responses (Figure 3 and Figure 4), which were correlated with reduced tumour numbers [15]. These observations suggest that pharmacological ROCK inhibition could have a similar effect as the ROCK1nc mutation in promoting tumour suppressive immune responses, which could result in a novel clinical use for ROCK inhibitors as pro-immunogenic agents for cancer therapy.

To test this possibility, 14 day old male ROCK1wt mice were IP injected with 400 µg DEN, and then at 10 weeks mice were orally administered either with water vehicle or 2 mg of the ROCK inhibitor fasudil [19] twice daily until mice reached 41 weeks (Figure 5A). Fasudil treatment had no significant effects on whole body, liver, lung or spleen weights (Appendix A). In addition, fasudil treatment had no significant effects on blood haemoglobin levels or the numbers of circulating platelets, white cells, lymphocytes or monocytes, but did result in a reduction in neutrophil numbers (Appendix A), and no effect on biochemical markers of liver health and function except for decreased serum levels of aspartate aminotransferase (Appendix A). While liver tumours were observed in all mice for both treatment groups (Figure 5B), tumour numbers (Figure 5C), total tumour areas (Figure 5D), and total tumour volumes (Figure 5E) were significantly lower in the fasudil treated mice. There were no significant differences in S100A9^+^ neutrophil (Figure 5F and Appendix A) or CD8^+^ T cell numbers (Figure 5G) in liver tumours without or with fasudil treatment. To corroborate these observations, a comparable experiment substituted the structurally unrelated ROCK inhibitor AT13148 [20,21,22] for fasudil. ROCK1wt mice that had been injected IP at 14 days with 400 µg DEN were then orally administered three times per week with either vehicle (10% DMSO, 1% Tween 20 in water) or 40 mg/kg AT13148 until mice reached 41 weeks (Figure 6A). AT13148 treatment had no significant effects on whole body, liver, lung or spleen weights (Appendix A). In addition, AT13148 had no significant effect on blood haemoglobin levels or the numbers of circulating platelets, lymphocytes, monocytes or neutrophils, although white cell numbers were significantly decreased (Appendix A). AT13148 had no significant effects on biochemical markers of liver health and function (Appendix A). All livers from both treatment groups had tumours (Figure 6B), although there were fewer tumours (Figure 6C), smaller tumour areas (Figure 6D), and tumour volumes (Figure 6E) for the AT13148 treated mice. There were no significant differences in neutrophil numbers between the treatment groups (Figure 6F), but AT13148 treatment did result in statistically significantly fewer CD8^+^ T cells in liver tumours (Figure 6G). Although both ROCK inhibitors fasudil and AT13148 had similar anti-tumour effects (Figure 5 and Figure 6), neither treatment increased neutrophil nor CD8^+^ T cell recruitment to tumours in contrast to the observations in ROCK1nc mice (Figure 3 and Figure 4).

To characterize the effect of systemic ROCK inhibition on immune cell recruitment to the liver in response to acute DEN treatment, 10 week old male ROCK1wt mice were orally administered three times per week with either vehicle or 40 mg/kg AT13148 for 10 days, with all mice receiving IP injections of 100 µg DEN on the 7th day. The addition of AT13148 with DEN did not result in significant differences in whole body or liver weight, or in blood haemoglobin levels or numbers of circulating white cells, monocytes or neutrophils relative to DEN treatment alone (Appendix A). However, AT13148 did significantly reduce the number of circulating platelets (Figure 7A) and lymphocytes (Figure 7B). Analysis of immune cell numbers in dissociated livers by flow cytometry revealed that AT13148 significantly reduced the number of dendritic cells (Figure 7C), NK cells (Figure 7D), CD8^+^ T cells (Figure 7E) and Kupffer cells (Figure 7F) relative to vehicle treated mice. The numbers of recruited macrophages, neutrophils, plasmacytoid dendritic cells, CD4^+^ T cells, Tregs, γδ T cells and NKT cells in DEN treated livers were not different between vehicle and AT13148 treated mice (Appendix A). These findings indicate that the ROCK inhibitor AT13148 had an overall negative effect on immune cell numbers in mouse livers following acute DEN treatment.

Previous studies showed that fasudil reduced the growth of hepatocellular carcinoma tumours driven by expression of oncogenic NRas plus over-expressed Akt [42], consistent with the observation that fasudil reduced the tumour number, area and volumes induced by DEN in Figure 5. This effect of fasudil was attributed to the induction of cell cycle arrest and not apoptosis [42] To determine if AT13148 worked via a similar mechanism, primary hepatocytes were isolated and treated with varying doses of AT13148 from 6.25 to 400 µM, and then cells were observed by time-lapse microscopy for up to 20 h. By counting the number of cells that were positive for green fluorescence when incubated with CellEvent caspase3/7 fluorogenic substrate, time and concentration dependent caspase activation was detected, peaking 3–5 h after the start of incubation (Figure 8A, left graph). Dose–response analysis at the 5 h time point yielded an EC_50_ = 256 µM (Figure 8A, right graph). Quantification of dead cells using SYTOX green, which become fluorescent after crossing the permeable plasma membrane of dead cells and binding to nucleic acids, indicated that cell death largely plateaued after 10–12 h at higher AT13148 concentrations (Figure 8B, left graph), with dose–response analysis after 20 h producing an EC50 = 22.3 µM. These results suggest that, in contrast to fasudil, the anti-tumour effects of AT13148 are likely induced, at least in part, through the induction of apoptosis. It should be taken into consideration that the isolation and culturing of primary hepatocytes results in changes to their metabolism that might lead to differences with in vivo responses to drug treatments [43].

## 4. Discussion

One of the hallmarks of cancer is the ability to evade destruction by the immune system [44]. Immunoevasion is accomplished by a variety of means, but the strategy of reactivating the immune system as an cancer therapeutic strategy has largely focussed on blocking the checkpoint inhibitor PD-1/PD-L1 and CTLA-4 pathways to enable tumour-reactive T cells to muster effective antitumor responses [45,46,47]. An additional strategy that is still in the process of becoming widely adopted is to intentionally induce ICD to stimulate the immune system in concert with conventional targeted therapies [8]. The effect of ICD is to promote the action of dying cells as immunological adjuvants through the release of DAMPs that promote the recruitment and activation of antigen-presenting cells, ultimately resulting in activation of cytotoxic T lymphocyte-mediated adaptive immunity and the establishment of immunological memory of tumour antigens and neoantigens [6]. Cancer therapies are typically developed through an iterative process that includes testing on established tumour cell lines in vitro and xenograft tumours grown in immunocompromised mice to examine their anti-cancer effects, which would not necessarily include assessments of their potential immunostimulatory or immunosuppressive effects [48]. Nevertheless, some commonly used cancer chemotherapies do in fact induce ICD, including bleomycin, bortezomib, doxorubicin, and oxaliplatin, although whether immunostimulation is a meaningful component of their therapeutic mechanisms of action is not clear [48].

ROCK1nc mice express a mutated form of ROCK1 in which the caspase cleavage site DETD (amino acids 1110–1113) was changed to DETA, making the protein resistant to caspase cleavage [12,15]. The apoptotic death of ROCK1nc cells is characterized by lower MLC phosphorylation, reduced cellular contractile force, altered morphologies and greater HMGB1 release relative to cells expressing wild-type ROCK1 [15]. In ROCK1nc mice, treatment with the hepatoxin DEN resulted in greater liver damage and neutrophil recruitment than in ROCK1wt mice, both of which were reduced through inhibition of HMGB1 or its receptor TLR4 [15]. In this study, acute DEN-induced liver damage was associated with greater induction of pro-inflammatory cytokines and chemokines in ROCK1nc mice compared to ROCK1wt mice (Figure 2D). Taken together, these results are consistent with cell death in ROCK1nc cells being immunogenic. In the long term, DEN-induced fewer HCC tumours in ROCK1nc mice [15], with more high grade steatosis (Figure 3A), and more neutrophils (Figure 3B) and CD8^+^ T cells (Figure 4B) in ROCK1nc tumours relative to ROCK1wt tumours. Levels of F4/80^+^ macrophages (Figure 3C) and Foxp3^+^ Treg cells (Figure 4C) were not significantly different between the genotypes. A caveat to the DEN-induced mouse HCC model is that its etiology may differ from human HCC, which is associated with chronic inflammation in many patients. [49] Clinical studies in human HCC patients have previously determined that high CD8^+^ T cells and/or high CD8^+^/Foxp3^+^ ratios are associated with increased overall and disease-free survival of HCC patients [36,37,38], suggesting that ICD in ROCK1nc tumours, which resulted in high CD8^+^ T cell numbers and CD8^+^/Foxp3^+^ ratios, induced tumour-suppressive immunity. From a clinical perspective, the obvious question is whether pharmacological inhibition of ROCK activity would provoke similar phenotypes.

Two structurally unrelated ROCK inhibitors, fasudil and AT13148, were selected due to their previous efficacy in mouse tumour studies [22,24] and because both have been shown to be safe for administration to human patients. Fasudil has been approved and safely used in Japan and China for many years for the treatment of cerebral vasospasm associated with subarachnoid hemorrhage [50], and there have been additional proposed applications of fasudil as a therapy for neurodegenerative diseases such as Amyotrophic Lateral Sclerosis [51,52] and cardiovascular diseases [53]. Development of AT13148 reached Phase I testing in patients with solid tumours [21]. Both fasudil and AT13148 reduced HCC tumour numbers, areas, and volumes (Figure 5C–E and Figure 6C–E) indicating beneficial therapeutic actions, but neither led to increased numbers of neutrophils or CD8^+^ T cells in tumours relative to vehicle controls (Figure 5F–G and Figure 6F–G). In fact, AT13148 significantly reduced CD8^+^ T cells in HCC tumours (Figure 6F). While the ROCK1nc mutation only inhibits activation by caspase cleavage in apoptotic cells, pharmacological inhibitors block activation resulting from caspase cleavage and RhoA binding, as well as reducing basal kinase activity, of both ROCK1 and ROCK2 in all cells in the body [15,20,54]. The actions of ROCK inhibitors that exceed the effect of the ROCK1nc mutation on kinase activity, combined with the widespread targeting of apoptotic and non-apoptotic cells, likely accounts for the differences observed for neutrophil and CD8^+^ T cells in HCC tumours between the genetic ROCK1nc modification and pharmacological ROCK inhibition. Given the significant inhibitory effect of AT13148 on CD8^+^ T cell numbers in HCC tumours, the effects on immune cell numbers in livers in response to acute DEN damage was examined. There were significantly fewer circulating platelets and lymphocytes in AT13148 treated mice relative to vehicle controls (Figure 7A,B), while the numbers of dendritic, NK and CD8^+^ T cells observed in ROCK1wt livers in response to DEN treatment were reduced (Figure 7C–E). These observations are consistent with the reduction in CD8^+^ T cells found in HCC tumours in AT13148 treated mice (Figure 6F).

Apoptotic ROCK1nc cells released more of the DAMP protein HMGB1 and lactate dehydrogenase (LDH), a cytosolic protein often used as a surrogate marker of cell death [15], indicating that their cell death was more necrotic-like than the typical apoptotic death of ROCK1wt cells. We previously showed that pharmacological ROCK inhibition during early apoptosis reduced the release of LDH and the DAMP proteins Histones H1.2, H2A, H2B, H3 and H4, S100-A4 and S100-A11, and heat shock proteins HSPA4, HSPA5, HSP8, HSP90AA1, HSP90B1 and HSP90B2 relative to control apoptotic cells [16]. The reduced release of cellular contents in ROCK inhibitor treated cells during early apoptosis was attributed to decreased production of apoptotic bodies, which do not consistently re-form impermeable membranes after detaching from apoptotic cell bodies, thus allowing for leakage of their contents. As a result, a potential explanation for the differences in neutrophil and CD8^+^ T cell numbers in HCC tumours could be due to there being more DAMP release in ROCK1nc mice than in ROCK inhibitor treated mice. Additional possibilities are that due to the central role of ROCK signalling in cell motility [55], pharmacological ROCK inhibition impaired the ability of neutrophils and CD8^+^ T cells to migrate towards attractant stimuli, or that systemic ROCK inhibitor administration affected immune cell activation and responsiveness.

## 5. Conclusions

Although the physical disruption of cells by irradiation, photodynamic therapy and high hydrostatic pressure have been shown to induce ICD and could be used in combination therapies to stimulate anti-tumour immune responses, altering the morphological changes associated with apoptotic cell death by pharmacologically blocking ROCK activity is not likely to be therapeutically beneficial. The ability of ROCK inhibitors to reduce HCC tumour numbers, areas and volumes (Figure 5C–E and Figure 6C–E) may be related to their effects on blocking invasive tumour growth [22,24]. When the ability of ROCK inhibitors to improve the efficacy of therapeutic agents by increasing tumour penetration [23,56] is also considered along with their anti-metastatic properties [57], then there may still be clinical benefits from the inclusion of ROCK inhibitors in HCC combination therapies.

## Figures and Tables

**Figure 1 cancers-14-05943-f001:**
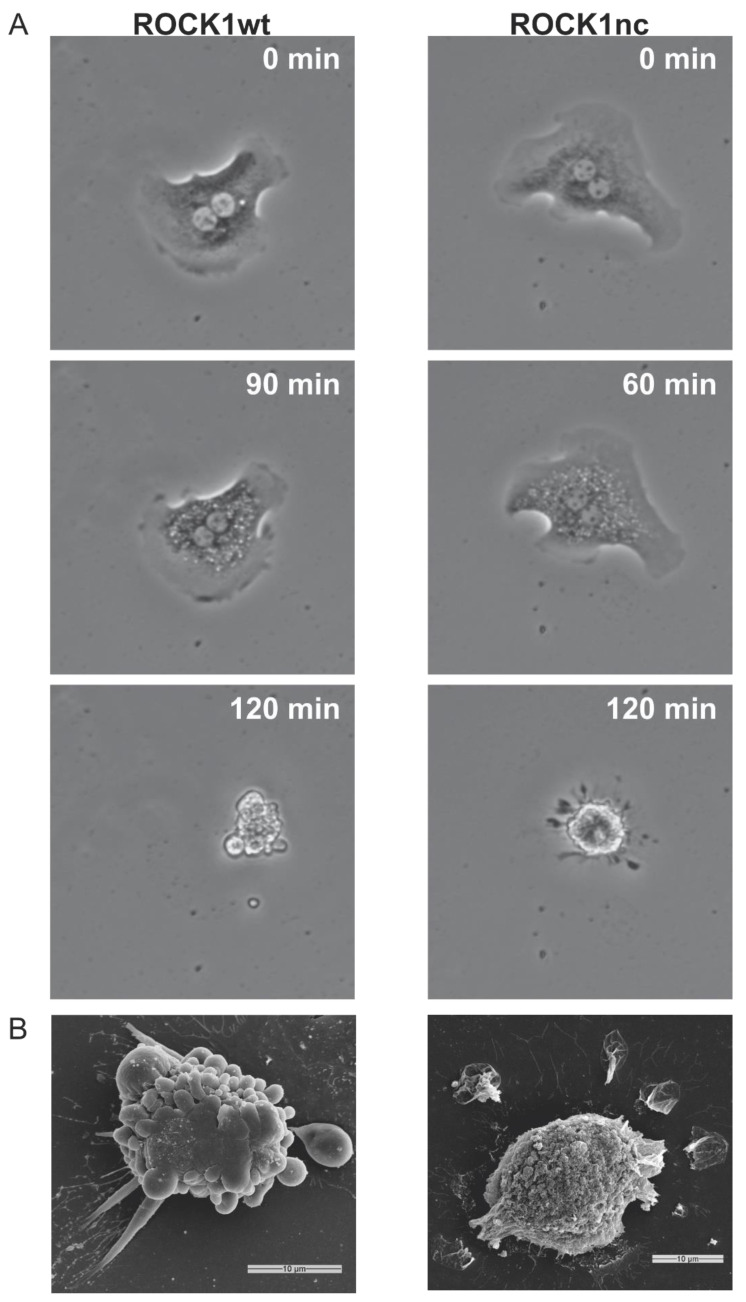
ABT-199 induces a distinct apoptotic morphology in ROCK1nc hepatocytes. (**A**) Representative images of primary hepatocytes isolated from ROCK1wt (left panels) and ROCK1nc (right panels) mice and treated with 1 µM ABT-199 to induce apoptosis. (**B**) Representative scanning electron micrographs of ROCK1wt (left) and ROCK1nc (right) hepatocytes. Scale bars = 10 µm.

**Figure 2 cancers-14-05943-f002:**
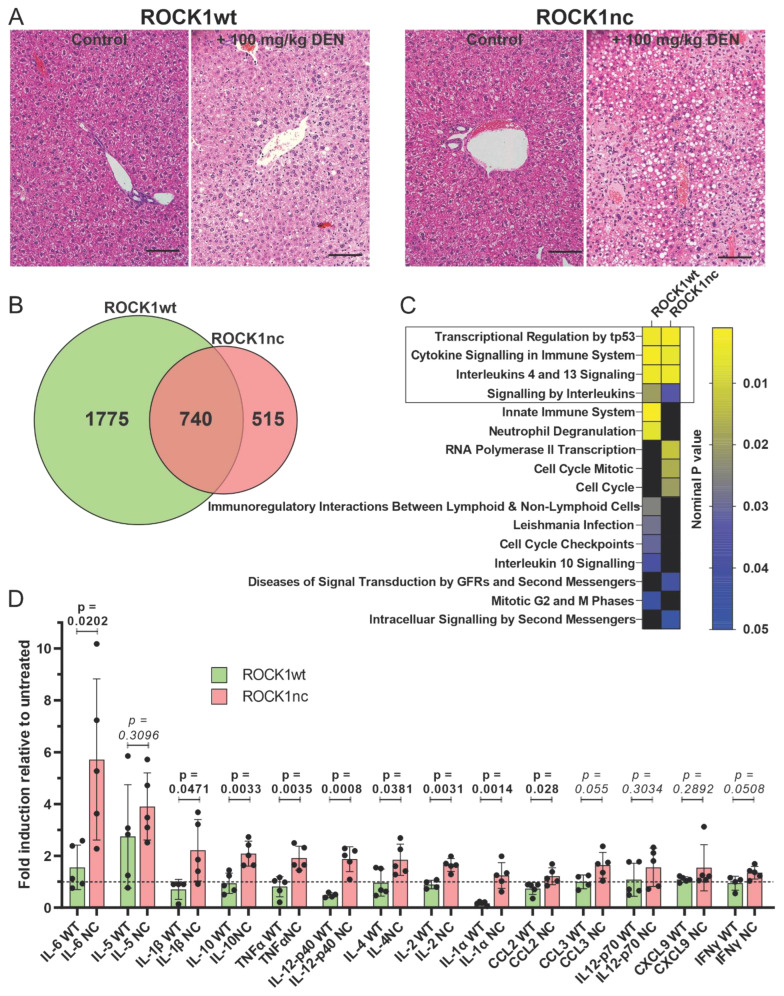
Acute DEN treatment induces steatosis and cytokine/chemokine signalling in ROCK1nc primary hepatocytes. (**A**) Haematoxylin and eosin (H&E) staining of representative liver sections from control ROCK1wt or ROCK1nc mice (left panels) or 72 h after DEN injection (right panels). Scale bars = 100 μm. (**B**) Venn diagram showing the number of genes significantly (adjusted *p* < 0.05) changed in expression by at least 1.5 fold in ROCK1wt and ROCK1nc livers 72 h after DEN injection. (**C**) Reactome gene sets significantly associated (nominal *p* < 0.05) with genes that were significantly (adjusted *p* < 0.05) altered in expression by at least 1.5 fold in ROCK1wt or ROCK1nc livers 72 h after DEN injection. (**D**) Induction of cytokines/chemokines 72 h after DEN injection relative to untreated ROCK1wt and ROCK1nc mice. Means ± standard deviation, *n* = 5 independent determinations. Unpaired Student’s *t*-test, where *p* values less than 0.05 are indicated in bold typeface.

**Figure 3 cancers-14-05943-f003:**
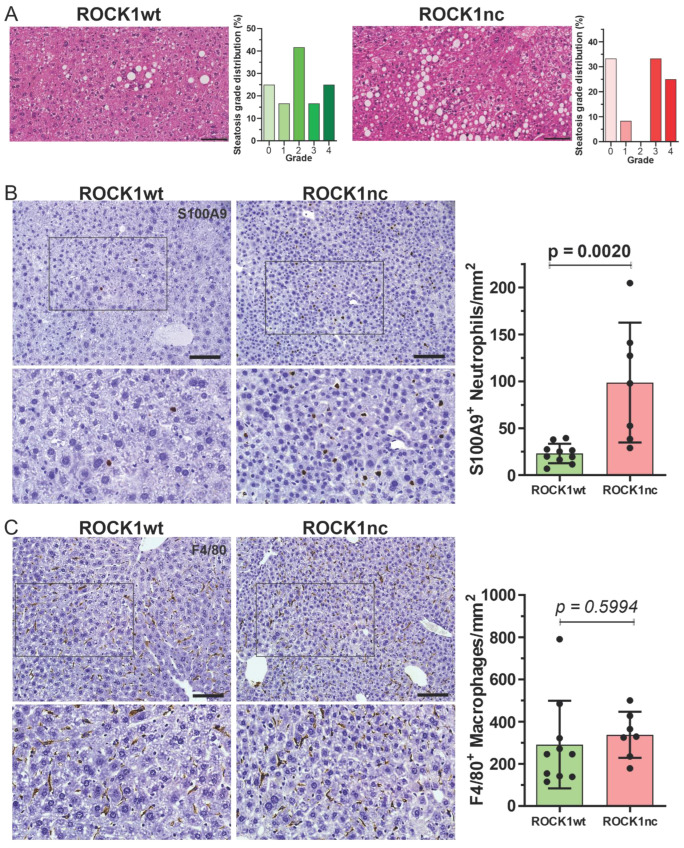
ROCK1nc liver tumours show greater steatosis and neutrophil numbers than ROCK1wt liver tumours. (**A**) H&E staining of representative liver tumour sections from ROCK1wt or ROCK1nc mice 9 months after DEN treatment. Steatosis was determined on H&E-stained sections as a percentage of the total area from 10 random fields per section per mouse, and graded according to the following criteria: grade 0, <5%; grade 1, 5–25%; grade 2, 26–50%; grade 3, 51–75%; grade 4, >75%. Scale bars = 100 µm. The example ROCK1wt field was grade 2, and the ROCK1nc field was grade 4. (**B**) Representative Immunohistochemical (IHC) images of anti-S100A9 antibody-stained formalin-fixed paraffin embedded (FFPE) liver sections showing neutrophils (brown) in liver tumours (left panels), and corresponding bar graph (right panel). S100A9 positive neutrophils were scored as number per mm^2^ in 5 random fields per section per mouse. (**C**) Representative IHC images of anti-F4/80 antibody-stained liver sections showing macrophages numbers in liver tumours (left panels), and corresponding bar graph (right panel). F4/80 positive macrophages were scored as number per mm^2^ in 5 random fields per section per mouse. Scale bars = 100 µm. Means ± standard deviation, ROCK1wt *n* = 10 mice, ROCK1nc *n* = 7 mice. Unpaired Student’s *t*-test, *p* values less than 0.05 indicated in bold typeface.

**Figure 4 cancers-14-05943-f004:**
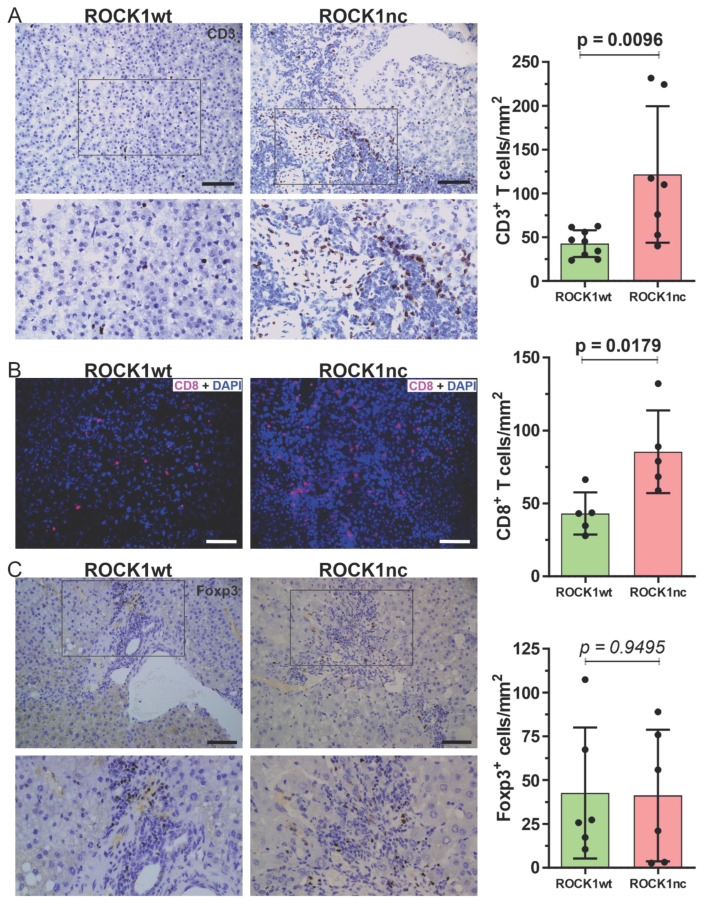
ROCK1nc liver tumours show greater CD3+ and CD8+ T cell numbers than ROCK1wt liver tumours. (**A**) Representative IHC images of anti-CD3 antibody-stained liver sections showing CD3^+^ T cells (brown) in liver tumours (left panels), and corresponding bar graph (right panel). CD3^+^ positive T cells were scored as number per mm^2^ in 5 random fields per section per mouse. (**B**) Representative immunofluorescence images of anti-CD8 antibody-stained liver sections showing CD8^+^ T cells (red) and DAPI stained cells (blue) in ROCK1nc tumours (left panels), and corresponding bar graph (right panel). CD8^+^ T cells (red) were scored as numbers per mm^2^ in 5 random fields per section per mouse (**C**) Representative IHC images of anti-Foxp3 antibody-stained liver sections showing Foxp3^+^ Treg cells (brown) in liver tumours (left panels), and corresponding bar graph (right panel). Foxp3^+^ Treg cells were scored as number per mm^2^ in 5 random fields per section per mouse. Scale bars = 100 µm. Means ± standard deviation, ROCK1wt *n* = 10 mice, ROCK1nc *n* = 7 mice. Unpaired Student’s *t*-test, *p* values less than 0.05 indicated in bold typeface.

**Figure 5 cancers-14-05943-f005:**
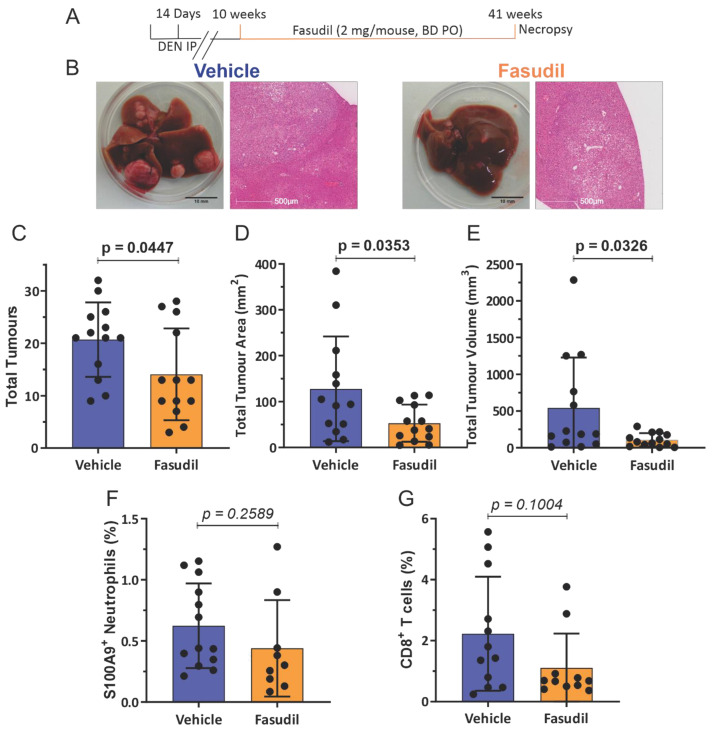
Fasudil-treated mice have fewer liver tumours but show no increase in immune cell recruitment. (**A**) Timeline of DEN + fasudil treatment regimen. (**B**) Macroscopic view of representative livers (scale bars = 10 mm) and representative H&E stained tumour sections (scale bars = 500 µm) from vehicle or fasudil treated mice. (**C**) Total tumour numbers in ROCK1wt mice treated with vehicle or fasudil. Means ± standard deviation, *n* = 13 mice per condition. (**D**) Total tumour areas in ROCK1wt mice treated with vehicle or fasudil. Means ± standard deviation, *n* = 13 mice per condition. (**E**) Total tumour volumes in ROCK1wt mice treated with vehicle or fasudil. Means ± standard deviation, *n* = 13 mice per condition. Liver cell suspensions were analyzed by flow cytometry to determine the relative proportions of (**F**) S100A9 positive neutrophils were scored as percentage of total cell counts per mouse. Means ± standard deviation, vehicle *n* = 13 mice, fasudil *n* = 9 mice, and (**G**). CD8^+^ T cells were scored as percentage of total cell counts per mouse. Means ± standard deviation, vehicle *n* = 12 mice, fasudil *n* = 11 mice. Unpaired Student’s *t*-test, *p* values less than 0.05 indicated in bold typeface.

**Figure 6 cancers-14-05943-f006:**
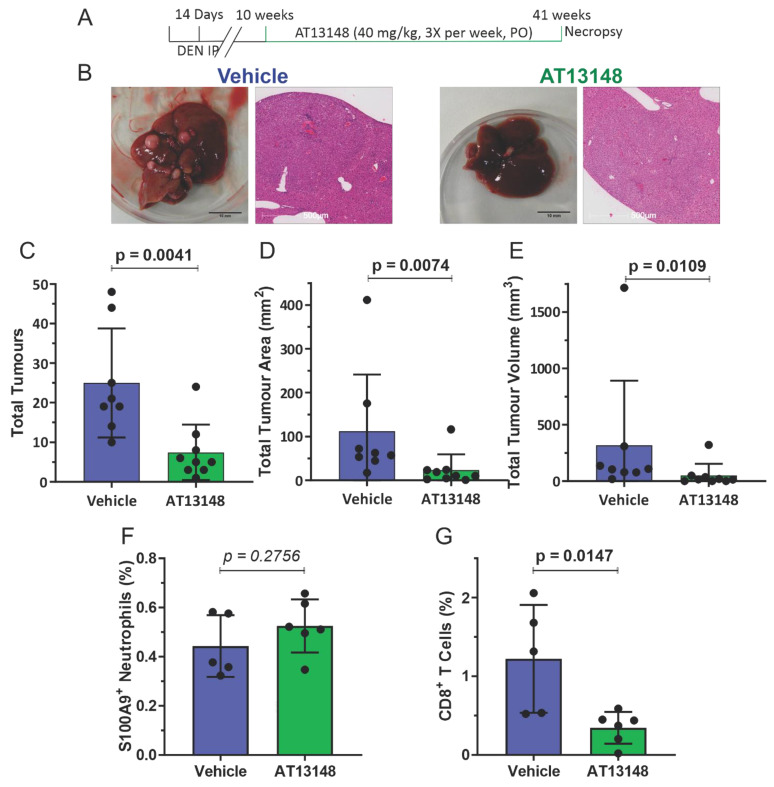
AT13148-treated mice have fewer liver tumours but show no increase in immune cell recruitment. (**A**) Timeline of DEN + AT13148 treatment regimen. (**B**) Macroscopic view of representative livers (scale bars = 10 mm) and representative H&E stained tumour sections (scale bars = 500 µm) from vehicle or AT13148 treated mice. (**C**) Total tumour numbers in ROCK1wt mice treated with vehicle or AT13148. Means ± standard deviation, vehicle *n* = 8 mice, At13148 *n* = 9 mice. (**D**) Total tumour areas in ROCK1wt mice treated with vehicle or AT13148. Means ± standard deviation, vehicle *n* = 8 mice, At13148 *n* = 9 mice. (**E**) Total tumour volumes in ROCK1wt mice treated with vehicle or AT13148. Means ± standard deviation, vehicle *n* = 8 mice, At13148 *n* = 9 mice. Liver cell suspensions were analyzed by flow cytometry to determine the relative proportions of (**F**) S100A9 positive neutrophils were scored as percentage of total cell counts per mouse. Means ± standard deviation, vehicle *n* = 5 mice, AT13148 *n* = 6 mice, and (**G**). CD8^+^ T cells were scored as percentage cell counts per mouse. Means ± standard deviation, vehicle *n* = 5 mice, AT13148 *n* = 6 mice. Unpaired Student’s *t*-test, *p* values less than 0.05 indicated in bold typeface.

**Figure 7 cancers-14-05943-f007:**
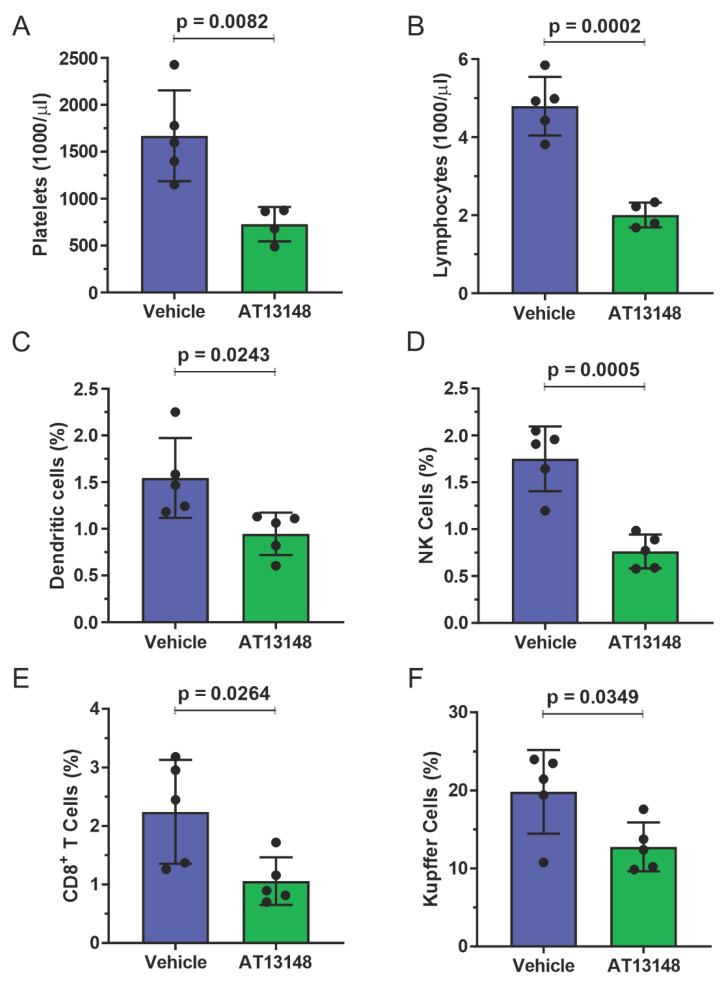
ROCK inhibitor AT13148 lowered circulating platelet and lymphocyte numbers, and reduced immune cell recruitment to livers following acute DEN treatment. ROCK1wt mice were treated with 40 mg/kg AT13148 or vehicle 3× per week via oral gavage for 10 days, then injected with 100 mg/kg DEN intraperitoneally 3 days prior to sacrifice to induce acute liver damage. Flow cytometry analysis of circulating (**A**) platelets and (**B**) lymphocytes in vehicle (*n* = 5) and AT13148 (*n* = 4) treated mice. Liver cell suspensions were analyzed by flow cytometry to determine the relative proportions of (**C**) dendritic cells, (**D**) natural killer (NK) cells, (**E**) CD8+ T cells and (**F**) Kupffer cells from vehicle and AT13148 treated mice (*n* = 5). Means ± standard deviation. Unpaired Student’s t-test, *p* values less than 0.05 indicated in bold typeface.

**Figure 8 cancers-14-05943-f008:**
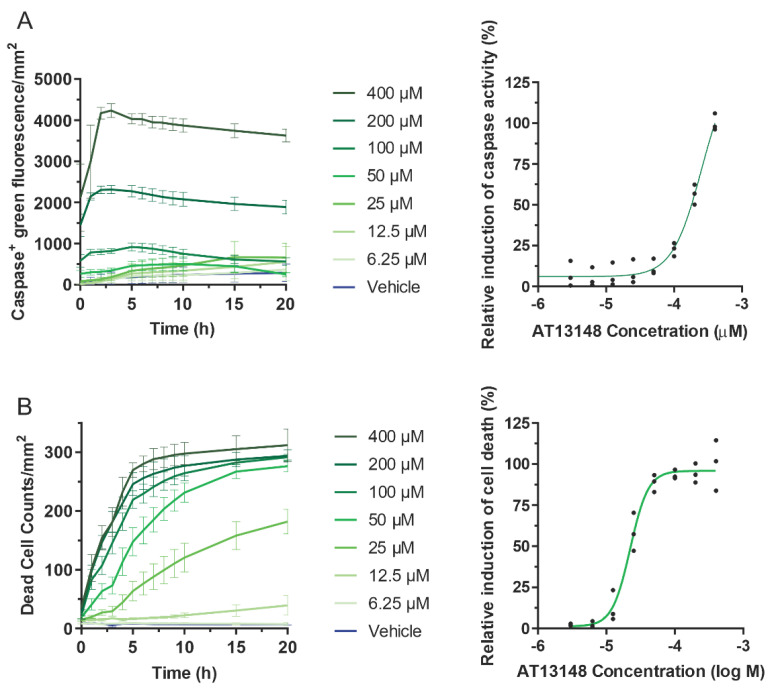
Primary hepatocyte apoptosis in response to AT13148 exposure. Time courses and dose responses of the numbers of (**A**) active caspase 3/7 positive and (**B**) dead hepatocytes to AT13148 concentrations ranging from 6.25-400 µM, over time. Caspase 3/7 activation was assessed with CellEvent green and apoptotic hepatocyte numbers were assessed with Sytox Green. Data shown are averaged results from 3 independent experiments. For the dose response time course graphs (left panels), lines represent means ± SD. For EC_50_ graphs (right panels), points represent normalized values from independent replicate experiments. Fitted lines were from non-linear regression fits with variable sloped to determine the EC_50_ values.

**Table 1 cancers-14-05943-t001:** Antigen retrieval methods, primary antibodies with dilutions, and incubation conditions for IHC. Citrate buffer (10 mM Sodium citrate, 0.05% (*v*/*v*) Tween 20). WB = water bath. RT = room temperature.

Antigen (Clone)	Species	Supplier (Catalogue Number)	Antigen Retrieval Solution	Retrieval Method	Antibody Dilution; Time & Temperature
CD3 (SP7)	Rabbit	Vector (VP-RM01)	Citrate buffer pH 6	WB; 98 °C; 25 min	1:75; 35 min, RT
F4/80 (Cl:A3-1)	Rat	Abcam (ab6640)	Proteinase K	10 min at RT	1:400; 35 min; RT
Foxp3 (FJK-16s)	Rat	eBioscience (14-5773-80)	Tris-EDTA buffer pH 9	WB; 98 °C; 20 min	1:50; O/N; 4 °C
S100A9 (M-19)	Goat	Santa Cruz (sc-8115)	Citrate buffer pH 6	WB; 98 °C; 25 min	1:1000; 35 min; RT
CD8 (4SM15)	Rat	eBioscience (14-0808-82)	Tris-EDTA buffer pH 9	WB; 98 °C; 20 min	1:500; 35 min; RT

**Table 2 cancers-14-05943-t002:** Liver flow cytometry staining panel.

Lymphoid Panel				Myeloid Panel			
Antigen (Clone)	Supplier	Dilu-tion	Fluoro-Phore	Antigen (Clone)	Supplier	Dilu-tion	Fluoro-Phore
B220 (RA3-6B2)	Biolegend	1:200	PerCPCy5.5	CD11b (M1/70)	Biolegend	1:200	BV570
CD3 (17A2)	Biolegend	1:200	FITC	CD11c (N418)	Biolegend	1:200	BV711
CD4 (RN4-5)	eBio-science	1:200	SB600	CD45 (30-F11)	Biolegend	1:200	BV510
CD8 (53-6.7)	Biolegend	1:200	PECy7	CD115 (AFS98)	Biolegend	1:100	BV421
CD19 (6D5)	Biolegend	1:200	PE	Clec4f	Biolegend	1:100	Goat/AF594
CD25 (PC61)	Biolegend	1:100	AF700	F4/80 (BM8)	Biolegend	1:100	Biotin/PE
CD45 (30-F11)	Biolegend	1:200	BV510	Ly6C (HK1.4)	Biolegend	1:400	PECy7
FoxP3 (MF14)	Biolegend	1:100	AF647	Ly6G (1A8)	Biolegend	1:200	AF700
NK1.1 (PK136)	Biolegend	1:200	BV711	MHCII (M5/114.15.2)	Biolegend	1:500	FITC
TCRβ (A57-597)	Biolegend	1:100	BV421	PDCA1 (927)	Biolegend	1:100	PerCPCy5.5
TCRγδ (GL3)	Biolegend	1:100	Biotin/AF594	Siglec H (551)	Biolegend	1:100	APC

## Data Availability

RNA sequencing data is available at: Olson, Michael; Julian, Linda (2022), “Liver RNAseq data”, Mendeley Data, V1, https://doi.org/10.17632/xk442frstc.1.

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
