# Peer review of "Immunogenic Death of Hepatocellular Carcinoma Cells in Mice Expressing Caspase-Resistant ROCK1 Is Not Replicated by ROCK Inhibitors"

_cancers, 2022, doi:10.3390/cancers14235943_

Round 1
Reviewer 1 Report
Here are my comments about this elegant article by Naylor et al. untitled: “Immunogenic Death of Hepatocellular Carcinoma Cells in Mice Expressing Caspase-Resistant ROCK1 is not Replicated by ROCK Inhibitors”:
- Very clear and well written work that is submitted by the authors. Moreover, it covers an important field and by reading carefully, I endorse publication of this work without minor revision on my side, so well done!
- As a general comment to the authors, would it be possible to add some FACS dot plots acquisition to the displayed figures in addition to the histograms?
- Also, in the Discussion part, the 2 first sentences are not enough referenced to my mind. Please refer to Zitvogel’s work by adding some papers from her group.
- Also, since authors have isolated hepatocytes, it would be noteworthy to state (1-2 sentences in the Discussion part) that during the isolation process of these cells, some alterations (especially the metabolism) can occur, and it may then not be the exact mirror of what happen in vivo. Please cite and refer to: Cassim S, Raymond VA, Lapierre P, Bilodeau M. From in vivo to in vitro: Major metabolic alterations take place in hepatocytes during and following isolation. PLoS One. 2017 Dec 28;12(12):e0190366. doi: 10.1371/journal.pone.0190366. PMID: 29284039; PMCID: PMC5746264.
Author Response
Reviewer 1
Here are my comments about this elegant article by Naylor et al. untitled: “Immunogenic Death of Hepatocellular Carcinoma Cells in Mice Expressing Caspase-Resistant ROCK1 is not Replicated by ROCK Inhibitors”:
- Very clear and well written work that is submitted by the authors. Moreover, it covers an important field and by reading carefully, I endorse publication of this work without minor revision on my side, so well done!
- As a general comment to the authors, would it be possible to add some FACS dot plots acquisition to the displayed figures in addition to the histograms?
RESPONSE: Contour plots were added in Supplemental Figure S5 from a representative experiment to illustrate how the flow cytometry was used to quantify immune cells.
- Also, in the Discussion part, the 2 first sentences are not enough referenced to my mind. Please refer to Zitvogel’s work by adding some papers from her group.
RESPONSE: The first two sentences in the Discussion refer to immunoevasion and checkpoint inhibitors. We have added to this part of the Discussion section the following review articles on line 693 that were authored or co-authored by Prof. Zitvogel: Clinical evidence that immunogenic cell death sensitizes to PD-1/PD-L1 blockade. Oncoimmunology. 2019 Jul 22;8(10):e1637188; Immune checkpoint inhibitors. J Exp Med. 2021 Mar 1;218(3):e20201979.
- Also, since authors have isolated hepatocytes, it would be noteworthy to state (1-2 sentences in the Discussion part) that during the isolation process of these cells, some alterations (especially the metabolism) can occur, and it may then not be the exact mirror of what happen in vivo. Please cite and refer to: Cassim S, Raymond VA, Lapierre P, Bilodeau M. From in vivo to in vitro: Major metabolic alterations take place in hepatocytes during and following isolation. PLoS One. 2017 Dec 28;12(12):e0190366. doi: 10.1371/journal.pone.0190366. PMID: 29284039; PMCID: PMC5746264.
RESPONSE: Thank you for highlighting this relevant reference it has been added to the final results section that relates to experiments with primary hepatocytes treated with AT13148. The following sentence has been added on lines 675-677: “It should be taken into consideration that the isolation and culturing of primary hepatocytes results in changes to their metabolism that might lead to differences with in vivo responses to drug treatments.”

Reviewer 2 Report
The article continues the author’s study published previously (Julian L. et al, eLife 2021; 10:e61983). The article is well written, experiments are well performed, and conclusions are supported by the obtained results. Here are few comments:
Major comments:
1. As shown in the Fig. 2B, there were 2-fold more genes aberrantly expressed following acute DEN treatment in the livers of ROCK1wt compared to ROCK1nc mice, whereas liver injury was more prominent in the ROCK1nc mice. This may indicate that many genes in the liver ROCK1nc have been already aberrantly expressed to some level, even at the steady state, before the treatment. Comparison of gene expression profiles of the untreated ROCK1wt and ROCK1nc mice may demonstrate this and reveal the impact of the ROCK1nc mutation on liver physiology at normal state, without any shock. This issue should be discussed.
2. The DEN model of HCC is not chronic inflammation-mediated, whereas in the patients, most HCC cases are chronic inflammation-mediated. The conclusions of the authors concerning the tested drugs would be much more relevant to clinical practice if they would perform the same experiments on a chronic inflammation-mediated rodent HCC model.
Minor comments:
1. ABT-199 concentration is shown 1 mM in the text (Results, 3.1.), but 1 mM in the legend to Fig. 1.
2. In the first paragraph of the section “Results 3.3.” (lane 561), the dot after the sentence “To test this possibility, 14 day old male.” is a typo, since the sentence continues at the next lane.
Author Response
Reviewer 2
The article continues the author’s study published previously (Julian L. et al, eLife 2021; 10:e61983). The article is well written, experiments are well performed, and conclusions are supported by the obtained results.Here are few comments:
Major comments:
- As shown in the Fig. 2B, there were 2-fold more genes aberrantly expressed following acute DEN treatment in the livers of ROCK1wt compared to ROCK1nc mice, whereas liver injury was more prominent in the ROCK1nc mice. This may indicate that many genes in the liver ROCK1nc have been already aberrantly expressed to some level, even at the steady state, before the treatment. Comparison of gene expression profiles of the untreated ROCK1wt and ROCK1nc mice may demonstrate this and reveal the impact of the ROCK1nc mutation on liver physiology at normal state, without any shock. This issue should be discussed.
RESPONSE: The reviewer has raised an interesting point. Our interpretation of the smaller number of significantly altered number of transcripts in the ROCKnc mice induced by DEN was that there was more chaos and randomness introduced because of the greater liver damage. By calculating Z score values from 5 replicate experiments for ROCK1wt and ROCK1nc mice treated with DEN for 17,557 transcripts with greater than 1.0 average sequence reads in both genotypes, frequency distribution analysis revealed that there were more extreme changes in the number of sequence reads in ROCK1nc mice. In fact, at all bin centres less than -1 and greater than 0.4, there are more Z scores from ROCK1nc mice than from ROCK1wt mice, reflecting greater variability in transcript numbers between experimental replicates. This greater variability made it less likely to achieve statistical significance when comparing data between untreated ROCK1nc to DEN-treated ROCK1nc than it was for comparisons between untreated ROCK1wt and DEN-treated ROCK1wt mice.
Nevertheless, we did compare the number of transcripts between untreated ROCK1wt and ROCK1nc mice, and determined that there were 571 from a total of 17,439 transcripts with greater than 1.0 average sequence reads for both genotypes that differed by >1.5 FC and adjusted p value < 0.05. Using the same Gene Set Enrichment Analysis of the Reactome gene sets that were used in Figure 2C, there were two gene sets that were significantly enriched. Neutrophil degranulation was positively enriched, with 12 genes expressed at higher levels in ROCK1nc mice than in ROCK1wt, and transport of small molecules was negatively enriched, with 9 genes expressed at lower levels in ROCK1nc mice than in ROCK1wt mice. The positive enrichment of neutrophil degranulation in ROCK1nc mice in basal conditions likely explains why it was not also positively enriched in this genotype when comparing untreated versus DEN treatment (Figure 2C). These observations suggest that even normal turnover of hepatocytes has an effect on neutrophil recruitment in ROCK1nc mice because of the more necrotic-like death these cells undergo. At this point, it’s not clear why there should have been a difference in expression of proteins involved in the transport of small molecules. Six members of the solute carrier family, aquaporin 4 and 8, and the mitochondrial calcium uniporter were all expressed at lower levels in ROCK1nc mice relative to ROCK1wt mice. The following sentences were added to the results section related to the RNA sequencing results in Figure 2 on lines 468-472. “Comparison of sequence reads from untreated ROCK1wt and ROCK1nc revealed that there were 572 genes differentially expressed by at least 1.5 fold (adjusted p < 0.05). Gene set enrichment analysis of the Reactome gene sets identified the gene set “Neutrophil degranulation” was significantly positively enriched (nominal p < 0.05) in ROCK1nc mice, and the gene set “Transport of small molecules” was significantly negatively enriched.”
- The DEN model of HCC is not chronic inflammation-mediated, whereas in the patients, most HCC cases are chronic inflammation-mediated. The conclusions of the authors concerning the tested drugs would be much more relevant to clinical practice if they would perform the same experiments on a chronic inflammation-mediated rodent HCC model.
RESPONSE: Thanks for suggesting this valid point, a new sentence in the Discussion has been added at lines 675-677 as follows: “A caveat to the DEN-induced mouse HCC model is that its etiology may differ from human HCC, which is associated with chronic inflammation in many patients.”
Minor comments:
- ABT-199 concentration is shown 1 mM in the text (Results, 3.1.), but 1 mM in the legend to Fig. 1.
RESPONSE: This typo has been corrected in the text.
- In the first paragraph of the section “Results 3.3.” (lane 561), the dot after the sentence “To test this possibility, 14 day old male.” is a typo, since the sentence continues at the next lane.
RESPONSE: Thanks for pointing out this typo, it has been corrected.

Reviewer 3 Report
This work from Gregory Naylor et al. describes the differences in immune cell infiltration and cell death in HCC between genetic models or inhibitors of ROCK1. The study is interesting and overall well done. Some aspects in my opinion can be improved to clarify some functions of the inhibitors in comparison to the genetic mouse model. See my comments below:
1. Are there any changes in the activation status of the leukocytes analyzed in ROCK1 inhibitors treated mice vs Ctrl?
2. Figure 1 A, why there are different time points between WT and nc (90 min and 60min)?
3. What are the levels of relevant cytokines in the plasma or in the tumor of ROCK1nc vs WT mice?
A similar analysis should be reported for the inhibitors treated mice.
HMGB1 levels should be also analyzed for the inhibitors treated mice.
4. Why report the % of neutrophils and CD8 T cells in Figures 5F and G? In the legend Authors mention that they were scored as number per mm2. For comparisons with Figure 4, it should be changed.
5. On a similar note, why is there so much sample number variability between apparently the same set of experiments (Figures 3 and 4) when analyzing Neutrophiles, CD3, CD8, or Foxp3 cells? Are there excluded samples from the analysis? if so, why? If exclusion criteria are used they must be reported.
Author Response
Reviewer 3
This work from Gregory Naylor et al. describes the differences in immune cell infiltration and cell death in HCC between genetic models or inhibitors of ROCK1. The study is interesting and overall well done. Some aspects in my opinion can be improved to clarify some functions of the inhibitors in comparison to the genetic mouse model. See my comments below:
- Are there any changes in the activation status of the leukocytes analyzed in ROCK1 inhibitors treated mice vs Ctrl?
RESPONSE: The activation status was not determined.
- Figure 1 A, why there are different time points between WT and nc (90 min and 60min)?
RESPONSE: The images in Figure 1A are representative screen grabs from the time lapse videos, the intermediate time points reflect the variability in the time course of contraction between individual cells. Importantly, the morphologies of the hepatocytes at the final 120-minute time point depict the differences between the WT and NC genotypes. The supplementary videos S1 and S2 that were submitted with the manuscript shows the morphologies of cell death over time.
- What are the levels of relevant cytokines in the plasma or in the tumor of ROCK1nc vs WT mice? A similar analysis should be reported for the inhibitors treated mice. HMGB1 levels should be also analyzed for the inhibitors treated mice.
RESPONSE: We agree that these might have been interesting experiments, but cytokine levels were not determined in the plasma or tumours of mice with HCC. Similarly, cytokine or HMGB1 levels were not analyzed in the inhibitor treated mice.
- Why report the % of neutrophils and CD8 T cells in Figures 5F and G? In the legend Authors mention that they were scored as number per mm2. For comparisons with Figure 4, it should be changed.
RESPONSE: Thank you for pointing out this typographical error. The proportions of neutrophils and CD8+ T cells were determined by flow cytometry in Figures 5F and 5G as we; as in Figures 6F and 6G, which is why the values were reported as percentages of cells counted. This has been corrected in the Figure 5 and 6 Legends.
- On a similar note, why is there so much sample number variability between apparently the same set of experiments (Figures 3 and 4) when analyzing Neutrophiles, CD3, CD8, or Foxp3 cells? Are there excluded samples from the analysis? if so, why? If exclusion criteria are used they must be reported.
RESPONSE: The varying sample numbers simply result from the fact that different stainings were done at different times, and the number of stained samples were not consistent. No samples that were stained and analysed were excluded, the results show all data acquired.

Round 2
Reviewer 3 Report
No more comments